# The Effects of Cu Powder on the Interface Microstructure Evolution of Hot-Rolled Al 6061/Mg M21/Al 6061 Composite Plates During Annealing

**DOI:** 10.3390/ma18030655

**Published:** 2025-02-02

**Authors:** Na Yang, Xianquan Jiang, Ruihao Zhang, Jian Li, Kaihong Zheng, Fusheng Pan

**Affiliations:** 1School of Materials and Energy, Southwest University, Tiansheng Road 2, Beibei District, Chongqing 400715, China; lindayang1314@163.com (N.Y.); zhangruihao95@163.com (R.Z.); lj267689494@163.com (J.L.); 2School of Mechanical Engineering, Guizhou University of Engineering Science, Xueyuan Road 1, Qixingguan District, Bijie 551700, China; 3Institute of New Materials, Guangdong Academy of Sciences, Changxing Road, Tianhe, Guangzhou 510650, China; zhengkaihong@gimp.gd.com (K.Z.); fspan@cqu.edu.cn (F.P.); 4College of Materials Science and Engineering, Chongqing University, Chongqing 400045, China

**Keywords:** Al-Mg composite plate, interface microstructure, annealing temperature, Cu powder

## Abstract

This study achieved the successful creation of a 6061/M21/6061 composite sheet, with Cu powder incorporated in the middle, through a two-pass hot roll bonding process. The effect of Cu powder addition on interface microstructure evolution of Mg-Al composite plate during annealing was studied. The results show that the incorporation of copper powder significantly suppresses the formation of Mg-Al intermetallic compounds (IMCs) at the boundary of Al-Mg bonded plates. The IMCs’ thickness of composite plate Mg-Al interface absent Cu powder increased from 7.0 µm at 250 °C to 61.2 µm at 400 °C, showing a rapid growth trend. On the contrary, in the area with Cu powder of composite plate containing Cu powder, when the temperature ranges from 250 °C to 350 °C, the Mg-Al diffusion layer is thin and only varies between 1 µm and 3.2 µm and, even when the temperature rises to 400 °C, the diffusion layer is only 18.8 µm. At a constant temperature, the diffusion rate of IMCs in the Cu powder-containing region of the composite plate is significantly lower than that in the region without Cu powder. Upon the addition of Cu powder, Al_2_Cu and Al_0.92_Cu_1.08_Mg phases are formed, which decrease the proportion of the brittle phases Al_3_Mg_2_ and Mg_17_Al_12_ at the composite plate interface, thereby effectively mitigating the diffusion of IMCs within the Mg-Al interface. This presents a novel concept for the investigation of enhanced interface bonding and the fabrication of Mg-Al composite plates.

## 1. Introduction

In recent years, the pursuit of lightweight materials has emerged as a significant trend in the development of metal structural materials. Developing cost-effective magnesium–aluminum composite materials is essential for replacing conventional metal structural materials, which can help address the current energy crisis and reduce environmental pollution [1,2]. Numerous methods exist for the preparation of Mg/Al composite plates, including compound casting [3,4,5], accumulative bonding [6,7,8], roll solid-liquid rolling [9,10], and clad-rolling [11,12,13]. Among these techniques, the hot rolling method stands out as relatively efficient and straightforward, which renders it a viable option for industrial applications However, regardless of the manufacturing method employed to produce Al/Mg bimetals, significant quantities of hard and brittle Al-Mg intermetallic compounds—specifically Al_12_Mg_17_ and Al_3_Mg_2_—exhibit widespread and continuous precipitation at the boundary between aluminum and magnesium [14].

When the intermetallic compound exhibits a diffuse distribution, it can effectively pin the interface and contribute to its strengthening. However, when these intermetallic compounds establish a continuous distribution at the interface phase, significant thermal mismatch may occur, adversely affecting the mechanical characteristics of the composite plates. Therefore, in the manufacturing process of Mg/Al composite plates, it is crucial not only to consider how to reasonably select the deformation heat treatment process but also to choose an appropriate intermediate implant layer.

Investigating the intermediate layer as an interface transition layer to establish a gradient distribution at the boundary can proficiently prevent the creation of hard and brittle IMCs. Preventing their rapid growth and ensuring a dispersed distribution represents a novel approach to interface strengthening. To elevate the interface’s bonding efficacy, various experts have encouraged the employment of an intermediate layer in the metal composites. Guo et al. [15] added a Zn layer in the middle of Mg/Al plate, and the plasticity of MgZn_2_ and ZnAl intermetallic compounds formed at the interface was relatively good, which multiplied the interface joining strength by a factor of two. However, the melting point of Zn was low (420 °C), and the hot rolling temperature was slightly higher, which was could cause burning and lead to automatic stripping of the interface. Liu et al. [16] inserted the Ni intermediate layer prepared by plasma spraying between Mg and Al matrix metals. After casting at 700 °C, the Ni intermediate layer was well combined with Mg-Al matrix. Compared with the composite plate prepared by directly using Mg/Al matrix, the interface strength was significantly improved. Jiang et al. [17] added a Zn interlayer casting in AZ91D and A356. The addition of the Zn interlayer restrained the formation of Mg-Al inter-metallics, but the shear strength of the Mg/Al bimetal was not high.

Previous studies have demonstrated significant progress in utilizing the intermediate layer as a transition layer at the Mg-Al interface. This advancement is primarily reflected in strategies such as incorporating an interlayer or coating the midsection of the Mg-Al plate, optimizing the rolling process, and investigating the formation of secondary phases with varying gradients to impede the development of hard and brittle phases—such as Al_3_Mg_2_—at the junction of magnesium–aluminum composite slabs, thereby enhancing interfacial bonding performance. However, reports on the effects of Cu powder on both mechanical properties and interfacial microscopic organization of Mg-Al alloy sheets remain scarce. The melting point of Cu reaches 1083 °C, making it more stable than Zn and less prone to combustion during high-temperature annealing. Additionally, Cu can react with Mg to form ternary phases such as Mg_2_Cu and Mg-Cu-Al, thereby preventing the creation of a sizeable diffusion layer formed by IMCs [18], which ultimately enhances the mechanical properties of the composite plate. In this study, Mg-Al composite panels were successfully fabricated by hot rolling an Al plate coated with a Mg plate and interspersed with Cu powder. The evolution of the interface microstructure in the Al/Mg/Al composite plate after annealing at various temperatures was systematically investigated, along with the effects of Cu powder injection on the interface diffusion layer of the Mg-Al composite plate. This study offers a novel viewpoint for reducing the development of fragile IMCs at the Al-Mg boundary and for crafting sophisticated Al/Mg/Al composite materials.

## 2. Experimental Procedures

### 2.1. Materials and Preparation of Al-Mg Composite Sheets

In the experiment, commercial M21 Mg and 6061Al plates, measuring by lengths of 200 mm and widths 100 mm by thickness of 1.0 mm, and Cu powder of approximately 75 μm, which was produced by China Metallurgical Xindun Alloys (Handan, China) were utilized. The coverage rate of Cu powder on the slab was 80%. Table 1 illustrates the chemical makeup of both the 6061 Al plate and the M21 Mg plate.

Figure 1 presents the schematic of the rolling process for the Al-Mg laminate sheet. The surfaces of the plates were scrubbed with steel wool and then cleaned with industrial alcohol. The plates were then stacked in preparation for the hot rolling of the Mg-Al composite metal sheets containing Cu powder, referred to as AP. The AP samples were created by artificially spreading Cu powder, resulting in non-uniform distribution between the Mg panel and Al plate. The region containing Cu powder within the interface diffusion layer is designated as the AP-with Cu powder area, while the remaining regions are referred to as the AP-without Cu powder area. For comparison purposes, a composite plate without Cu powder spread in the middle was also rolled, designated as NP. To stop the movement between varying metals during the roll joining procedure, the three sheets were clenched with rivets together to achieve pre-connection. After being held at an annealing temperature of 450 °C for 45 min in a heating furnace, the first rolling pass was conducted, decreasing the composite plate’s thickness from 3 mm to approximately 2 mm. The plates were then returned to the furnace for a heating hold of 5 min before proceeding with a second rolling pass. The composite plate ended up with a thickness of 1 mm, resulting in a decrease amounting to 66%.

After hot rolling, the samples were subjected to annealing treatments at temperatures ranging from 200 °C to 400 °C in 50 °C increments for one hour each. This study aims to investigate the effects of annealing on the evolution of microstructures and mechanical properties in the composite plates. For clarity, Table 2 summarizes the annealing parameters and their respective designations.

### 2.2. Characterization

The microstructure of the interfaces between Al and Mg, the widths of IMCs, and fracture morphology along the rolling direction of the specimens were characterized using scanning electron microscopy (SEM, Tescan Mira 3, Brno, Czech Republic) to assess element distribution via energy-dispersive X-ray spectroscopy (EDS), as illustrated in Figure 2. To qualitatively analyze the phase composition at the Mg/Al interface, the composite plate was sectioned along the Al/Mg interface depicted in Figure 2, following the hot rolling direction. Both stripping surfaces from the sides of Al and Mg were examined employing a Rigaku D/MAX-2500 X-ray diffractometer (XRD) (Tokyo, Japan). The interface of TC351 along the rolling direction was thinned using a focused ion beam (FIB, FEI Nova 450, Hillsboro, OR, USA) and subsequently examined by transmission electron microscopy (TEM, JEOL-2100F, Tokyo, Japan).

## 3. Results and Discussion

### 3.1. SEM and XRD Analysis of the Al/Mg Interface

Figure 3 illustrates the scanning electron microscopy (SEM) morphology of the interfaces in both AP and NP composite plates subsequent to being annealed at various temperatures for 1 h. Panels Figure 3a–e depict the interface characteristics of the AP composite sheet, while panels Figure 3f–j illustrate those of the NP multilayer plate. As shown in Figure 3a, after annealing at 200 °C, the interface bond appears smooth with no evident cracks, holes, or other defects, and there are no discernible diffusion layers. With an increase in temperature to 250 °C, minor zigzag pores begin to emerge at the Mg-Al interface along with a small quantity of diffusion layers, as observed in Figure 3b. However, from 300 °C to 400 °C, significant changes occur in the interface morphology of the AP composite plate. At an annealing temperature of 300 °C, a Cu powder implantation area is present on the boundary, causing the Mg-Al interface to bypass this region and diffuse along both sides of the Mg and Al plates. A small number of discontinuous holes appear near the Al side of the interface, as illustrated in Figure 3c. As the temperature continues to rise to 350 °C, the quantity of discontinuous holes near the Al side increases. In Figure 3d, it can be observed that in areas containing Cu powder particles at the interface, the diffusion layer at the Mg-Al interface becomes significantly thinner. Furthermore, Cu particulates directly create a barrier at the Al-Mg interface and impede the diffusion of Al and Mg elements across the interface.

At an annealing temperature of 400 °C, the diffusion of IMCs at the interface of the AP composite plate is impeded by the presence of Cu powder, resulting in a discontinuous diffusion state for Mg-Al IMCs that circumvents the Cu powder region. This is because, with the increase in temperature, the atomic motion of the atoms usually expands, promoting diffusion, but the presence of copper powder prevents or slows down the mutual diffusion between aluminum and magnesium. The copper powder may have changed the rate of IMC nucleation and growth, resulting in an uneven diffusion state at the interface. In areas where Cu powder is present, the width of the diffusion interface at the Mg-Al interface is reduced, with a few holes observed on the IMCs and uneven distribution of Cu within a white strip morphology. Additionally, some Cu particles are spalling off, leading to irregular holes on IMCs in regions containing Cu powder. This results from the varying diffusion rates of Mg and Al elements. Previous studies [19] have shown that the diffusion rate of aluminum atoms greatly exceeds that of Mg elements. Throughout the annealing treatment, Al atoms and Mg atoms diffuse together, forming an intermetallic phase at the Al-Mg junction. The distinct diffusion rates between Al and Mg atoms cause atoms to be lost adjacent to the Al side, resulting in Kirkendell holes.

In comparison, when the annealing temperature is set to 200 °C, the interface changes in the NP composite plate are similar to those observed in the AP composite plate, with no discernible diffusion layer present at the interface, as illustrated in Figure 3f. At an annealing temperature of 250 °C, Mg-Al diffusion layers begin to emerge at the interface; the bonding interface remains smooth and exhibits a thickness slightly greater than that of the AP composite plate, as shown in Figure 3g. As the annealing temperature increases from 300 °C to 350 °C, thin discontinuous faults emerge at the interface near the Al side, while the interface adjacent to the Mg side remains well bonded without any evident holes, cracks, or other defects. The interface bond is smooth, and the diffusion layer gradually thickens. As the temperature reaches 400 °C, the bonding morphology of the interface near the Mg layer transitions from a smooth bonding surface to a ‘convex’ structure with varying levels. Additionally, holes and a few discontinuous pores appear in the matrix near the Al side, along with a quick expansion of the diffusion layer thickness.

Table 3 presents the interface thicknesses of both AP and NP composite plates at various annealing temperatures for a duration of 1 h. As shown in Table 3, there is a significant difference in the thickness of the diffusion layer at the interface between the AP-with Cu powder area and the AP-without Cu powder area within the AP composite plate. The creation of Mg-Al IMCs begins with the interface temperature at 250 °C. At this temperature, the diffusion layer measures approximately 5.6 µm in the area without Cu powder. However, in the AP-with Cu powder region, the Mg-Al diffusion layer remains very thin, measuring only about 1.0 µm. As the temperature increases from 300 °C to 350 °C, in the AP-without Cu powder area, the thickness of Mg-Al IMCs rises from 13.7 µm to 18.5 µm. Interestingly, in contrast, the thickness of Mg-Al IMCs in the AP-with Cu powder region varies minimally, changing only from 1.9 µm to 3.2 µm. Even when reaching an annealing temperature of 400 °C, the thickness of the diffusion layer in regions containing Cu powder is merely 18.8 µm approximately.

In contrast, for NP composite plates without Cu powder, the thickness of IMCs increased from 7.0 µm at 250 °C to 61.2 µm at 400 °C. The Mg-Al IMCs in NP composite plates exhibited a rapid growth trend with increasing annealing temperature.

Figure 4 illustrates the changes in IMCs of both AP and NP composite plates as a function of annealing heat. Figure 4 indicates that the growth trend of IMCs in the AP-without Cu powder area is slightly lower than that observed in the NP composite plate with increasing temperature. However, in the AP-with Cu powder area, the growth rate of Mg-Al intermetallic compounds is considerably lower compared to the NP composite plates.

In summary, with an increase in annealing temperature, the interface near the Mg side of NP composite plates transitions from a smooth and flat structure to a ‘raised’ interface with varying heights. The thickness of IMCs at the Mg-Al interface of NP composite plates increased from 7.0 µm at 250 °C to 61.2 µm at 400 °C, demonstrating a rapid growth trend. In contrast, for AP composite plates, the bonding interface remains smooth and flat. However, discontinuities appear on the Al side starting at 300 °C. Notably, the Mg-Al IMCs at the bonding interface in the AP-without Cu powder area are significantly thicker than the IMCs in the AP-with Cu powder area. It is important to highlight that, in regions containing Cu powder within AP composite plates, when temperatures range from 300 °C to 350 °C, the Mg-Al diffusion layer is thinner—varying only between 1.9 µm and 3.2 µm—and even when reaching an annealing temperature of 400 °C, it rises only to approximately 18.8 µm. The presence of copper powder evidently hinders the rapid inter-diffusion of Mg and Al atoms. Thus, it inhibits the swift propagation of interfacial IMC layers.

The growth rate of the interface diffusion layer can be quantitatively expressed in relation to time using the following equation [20,21,22].(1)$$x=D\;{(t)}^{\frac{1}{2}}$$
where x represents the thickness of the region of diffusion, *D* denotes the speed of intermetallic compound development, and t signifies the annealing time [20,21,22]. Table 4 presents a detailed analysis of the growth rates of interface layers for both AP and NP composite plates at different annealing temperatures.

The correlation between the grain boundary diffusion coefficient (*D*) and annealing temperature (*T*) can be expressed as [23](2)$$D=D_{0}\exp\;(-\frac{Q}{RT})$$
where *D* is the growth rate of the IMCs, *D*_0_ is the exponential pre-factor, *Q* is the energy of activation, R is the gas constant (8.314 J/mol) and *T* is the absolute temperature [23].

Figure 5 illustrates the relationship between interface diffusion rate and temperature in AP-without Cu powder, AP-with Cu powder, and NP composite plates. As the temperature increases, the grain boundary diffusion coefficient (*D*) of both AP and NP composite plates exhibits an exponential increase with rising annealing temperatures. Consequently, the diffusion rate of these composite plates continues to rise. This phenomenon can be attributed to this phenomenon can be explained by enhancing elemental activity within the composite plate at elevated temperatures, which leads to a significant increase in the diffusion coefficient; this trend is particularly pronounced in NP composites. However, at equivalent temperatures, the diffusion rate of IMCs in regions containing Cu powder within AP is lower than that observed in NP composites. This reduction is attributed to the presence of implanted Cu powder, which interacts with Al and Mg during diffusion to form new phases; such phase formation results in higher activation energy for diffusion, leading to a decline in the diffusion rate [23]. Therefore, at the same temperature, the diffusion layer thickness of the AP-with Cu Powder area is significantly lower than that of the NP composite plate.

In conclusion, the interfacial morphology and growth rate of the diffusion layer at the interface of AP composite plates implanted with Cu powder exhibit significant differences compared to those of NP composite plates without Cu powder. These differences are closely associated with the novel phases formed at the interface of AP composite plates.

To more clearly observe the composition of the layer at the interface within the composite sheet, energy dispersive spectroscopy (EDS) line scanning was conducted on a sample of the AP composite plate annealed at 350 °C for 1 h, as illustrated in Figure 6. Figure 6 area A refers to the diffusion zone located close to the Mg portion, area B refers to the diffusion layer located adjacent to the Al side, and area C refers to the diffusion layer containing Cu powder. As shown in Figure 6a.

The area of material exchange is smooth and continuous, exhibiting significant thickness in areas without Cu powder. In contrast, in regions with Cu powder, the diffusion layer displays discontinuous variations with differing thicknesses. Figure 6b further clarifies that the diffusion layers of TC351 are primarily composed of two sublayers of Mg-Al IMCs in areas devoid of Cu powder. In contrast, within regions containing Cu powder, the interface composition predominantly consists of a polycrystalline structure formed by Mg-Al-Cu composite phases.

Table 5 shows the EDS data, point scanning analysis corresponding to the positions indicated in Figure 6, combined with an analysis of the Mg-Al binary phase diagram [13]. The possible phases at the interface are summarized in Table 5. The position labeled a1 is primarily composed of an Mg matrix. The main phase at position b1, located near the Mg side in line 1, is likely to be Mg_12_Al_17_, while the primary phase at position c1 could be Al_3_Mg_2_. Notably, Mg_12_Al_17_ forms a thinner diffusion layer compared to Al_3_Mg_2_. The phase identified at point a2 consists mainly of Mg-Al-Cu intermetallic compounds (IMCs). Point b2 predominantly contains Cu particles. It is clear that the thickness of Mg-Al IMCs is reduced here. The diffusion characteristics are illustrated in Figure 6d,e, where it can be observed that the interfacial diffusion layer in areas without Cu powder is significantly thicker than that in regions containing Cu powder, with measured thicknesses detailed in Table 3.

To investigate the interfacial phase makeup of AP and NP composite plates, XRD detection was performed on the Al and Mg sides of both types of plates annealed at different temperatures for 1 h. The sampling positions are illustrated in Figure 2, with results displayed in Figure 7. The XRD outcomes confirm the existence of an Al matrix at various annealing temperatures (1 h), as shown in Figure 7a–d. Additionally, the Al_3_Mg_2_ phase was detected on the peeling surface of both AP and NP composite plates starting from 300 °C. However, both Al_3_Mg_2_ and Al_12_Mg_17_ phases were observed on the Mg peel-off surface in these samples, as depicted in Figure 7a,c. This observation is attributed to the generation of the Al_3_Mg_2_ phase on the stripped-off Al side during the stripping process, which aligns with previous studies [17,24,25,26], indicating that an Al_12_Mg_17_ phase forms near the Mg side while an Al_3_Mg_2_ phase appears near the Al side. Furthermore, XRD analysis revealed Cu particles present on the Mg side. However, no significant Cu-Al-Mg compounds were detected, potentially due to limitations inherent to this detection method. To further investigate interface phases at varying annealing temperatures, transmission electron microscopy (TEM) will be employed for additional analysis.

### 3.2. TEM and EBSD Analysis of IMCs

To validate the SEM results presented in Figure 6 and the XRD findings shown in Figure 7, as well as to further investigate the interfacial phase composition depicted in Figure 6, TEM and EBSD were utilized to examine areas A, B, and C of Figure 6.

Figure 8 shows sample No.TC351 (i.e., the sample that underwent annealing annealed at 350 °C for 1 h), according to the FIB sampling location map of zones A, B and C and TEM results of each region in Figure 3. Region A is the sub-interface near Mg side, region B is the sub-interface near Al side, and region C is the diffusion layer containing Cu powder, which is the Mg-Cu-Al interface layer. High resolution and selective diffraction of the Mg/Al interface layers in region A indicate that the Al side of the composite plate is formed by solid solution and Al matrix, while the Mg side comprises an Mg matrix, which is consistent with the SEM results.

In the B area, at the Al-Cu interface, diffraction results indicated that an Al_2_Cu phase was present, as depicted in Figure 8d. This results from the fact that the atomic radius of Al is smaller than that of Cu, so the Cu occupies a large space in the lattice, and the barrier during diffusion is relatively small, so it is easier to diffuse to Al [27,28,29]; it combines with Al to form an Al_2_Cu phase, blocking most of the diffusion of Al to Mg and the Mg-Al-Cu interface is mainly formed in this region, so the diffusion layer in this region is relatively thin, at about 1.9 µm. Simultaneously, as the Cu element diffuses to both the Mg and Al sides, a Mg-Cu-Al interface is formed. The high-definition view of the Mg-Cu-Al interfacial structure and the corresponding selected area electron diffraction (SAED) pattern are presented in Figure 8, specifically in region C (i). The high-resolution results indicate that the sub-layer is predominantly composed of Al_2_Cu and Al_0.92_Cu_1.08_Mg phases, along with a minor presence of Phases with hardness and brittleness such as Al_3_Mg_2_ and Al_12_Mg_17_. This observation contrasts with the findings obtained from XRD analysis, primarily due to the limited quantity of Cu powder, which restricts its distribution at the interface and consequently diminishes the formation of phases in conjunction with Mg and Al. Moreover, XRD detection methods have inherent limitations that impede accurate identification of these two new phases, Al_2_Cu and Al_0.92_Cu_1.08_Mg.

The creation of two additional phases at the interface is attributed to the larger atomic radius of Cu compared to Al, which allows Cu to occupy a greater volume within the crystal lattice and results in a relatively lower diffusion barrier. Consequently, Cu can diffuse more readily into Al, facilitating the formation of the Al_2_Cu phase [23]. This process obstructs the distribution of most Al into Mg, leading to the development of hard and brittle phases, such as Al_3_Mg_2_ and Al_12_Mg_17_.

Figure 9 shows the inverse pole figure illustrations of AP and NP composite plates following annealing at 350 °C for 1 h. It is obvious that this annealing treatment results in equiaxed grains with fine particles on the magnesium side of the AP composite plate, exhibiting an average size of 3.26 μm, as exhibited in Figure 9b. By contrast, on the aluminum side, elongated grains are observed along with a significant variation in grain size, yielding an average particle size of 17.63 μm. Under the same conditions, the NP composite plate exhibits a similar grain shape on the Mg side but with a larger mean grain size of 7.34 μm, as shown in Figure 9c. The Al side also displays elongated grains accompanied by smaller grains, yielding an average particle size of 17.8 μm, as illustrated in Figure 9d. The grain sizes on both the Al and Mg sides of the AP composite plate are smaller than those observed in the NP composite plate. In contrast, the IMC interface consists of elongated and equiaxed mixed-phase grains within the AP composite plate, primarily composed of Al_0.92_Cu_1.08_Mg, Al_2_Cu, Al_3_Mg_2_, and Mg_17_Al_12_ phases, consistent with TEM results. The total proportions of the Al_0.92_Cu_1.08_Mg phase and Al_2_Cu phase are 6.1%, while the volume ratio of Al_3_Mg_2_ to Mg_17_Al_12_ is 1.2%. In the AP composite plate, the formation of new phases, such as Al_0.92_Cu_1.08_Mg and Al_2_Cu, reduces the volume fraction of brittle phases, like Al_3_Mg_2_ and Mg_17_Al_12_.

## 4. Conclusions

In our research, a Cu powder-reinforced Mg M21/Al 6061 combination plate was successfully fabricated through hot rolling. The influence of incorporating Cu powder on the interfacial structure of the Mg M21/Al 6061 composite plates was extensively analyzed. Based on the results, we can conclude as follows:(1)As the annealing temperature increases, the interface near the Mg section within the NP composite plate transitions from a smooth and flat structure to a ‘raised’ interface configuration with varying heights. In contrast, the interfacial microstructure of the AP composite plate evolves from a state characterized by minimal defects, such as pores and cracks, to a discontinuous interface configuration.(2)The width of Mg-Al IMCs at the Mg-Al interface of the NP composite plate increased from 7.0 µm at 250 °C to 61.2 µm at 400 °C, demonstrating a rapid growth trend. In contrast, in the AP area with Cu powder, when the temperature ranges from 250 °C to 350 °C, the Mg-Al diffusion layer remains thin, varying only between 1 µm and 3.2 µm and, even as the annealing temperature rises to 400 °C, this diffusion layer increases to only 18.8 µm. The incorporation of copper powder significantly suppresses the emergence of Mg-Al IMCs at the junction of Al-Mg composite plates.(3)The diffusion rates of the AP composite plate and NP composite plate increase with rising annealing temperatures. At an identical temperature, the dispersion rate of IMCs in the AP area containing Cu powder is significantly lower than that in the NP composite plate without Cu powder.(4)As the annealing temperature reaches 350 °C, the phases present at the interface in the NP composite plate devoid of Cu powder are predominantly brittle Al_3_Mg_2_ and Mg_17_Al_12_ phases. In contrast, the interfacial phases of the AP composite slab containing Cu powder consist mainly of Al_3_Mg_2_, Mg_17_Al_12_, Al_2_Cu, and Al_0.92_Cu_1.08_Mg phases. Specifically, the total proportion of the Al_2_Cu phase and Al_0.92_Cu_1.08_Mg phase is 6.1%, while the volume ratio of Al_3_Mg_2_ to Mg_17_Al_12_ is 1.2%. With the addition of Cu powder, new phases such as Al_2_Cu and Al_0.92_Cu_1.08_Mg are generated, which effectively reduce the proportion of brittle phases like Al_3_Mg_2_ and Mg_17_Al_12_ at the composite plate interface, thereby significantly diminishing IMC diffusion in the Mg-Al interface.

## Figures and Tables

**Figure 1 materials-18-00655-f001:**
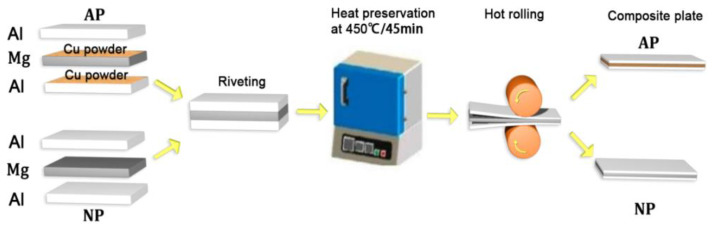
Process of hot rolling Al-Mg composite plate.

**Figure 2 materials-18-00655-f002:**
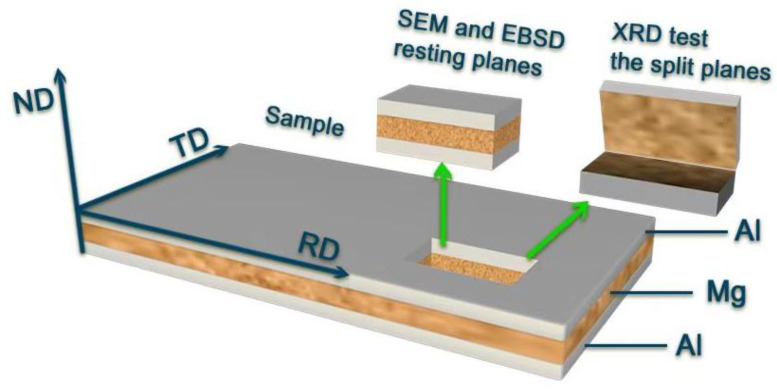
The sampling map of the characterized sample.

**Figure 3 materials-18-00655-f003:**
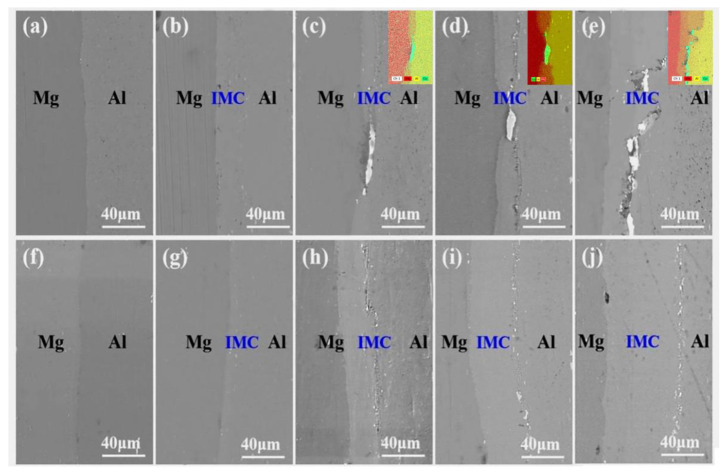
The SEM diagram of the interface between AP and NP at different annealing. Temperatures for 1 h. (**a**) TC201, 200 °C, (**b**) TC251, 250 °C, (**c**) TC301, 300 °C, (**d**) TC351, 350 °C, (**e**) TC401, 400 °C, (**f**) TN201, 200 °C, (**g**) TN251,250 °C, (**h**) TN301,300 °C, (**i**) TN351, 350 °C, (**j**) TN401, 400 °C.

**Figure 4 materials-18-00655-f004:**
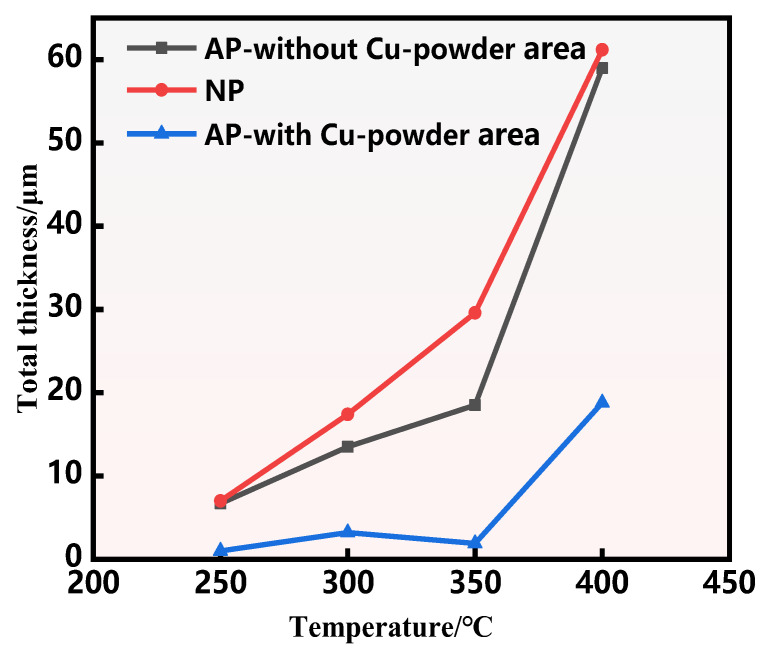
Influence of annealing temperature on the change in thickness of intermetallic compounds at the interface between AP and NP after 1 h of annealing.

**Figure 5 materials-18-00655-f005:**
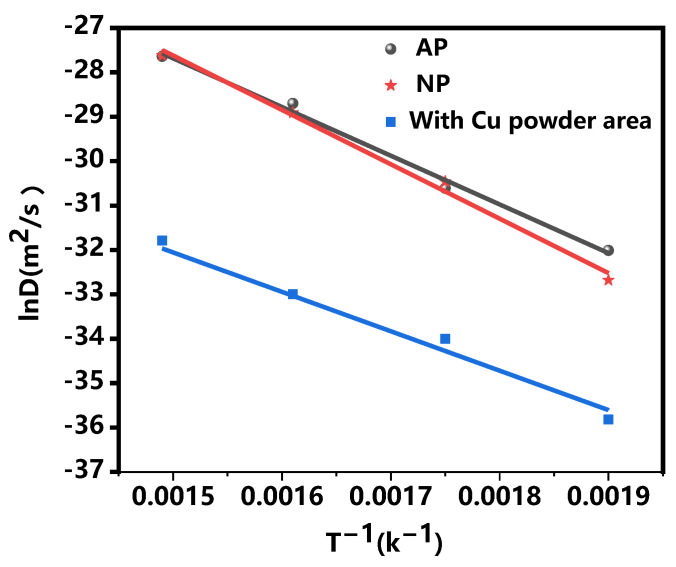
Al-Mg IMC expansion rates in relation to annealing temperatures.

**Figure 6 materials-18-00655-f006:**
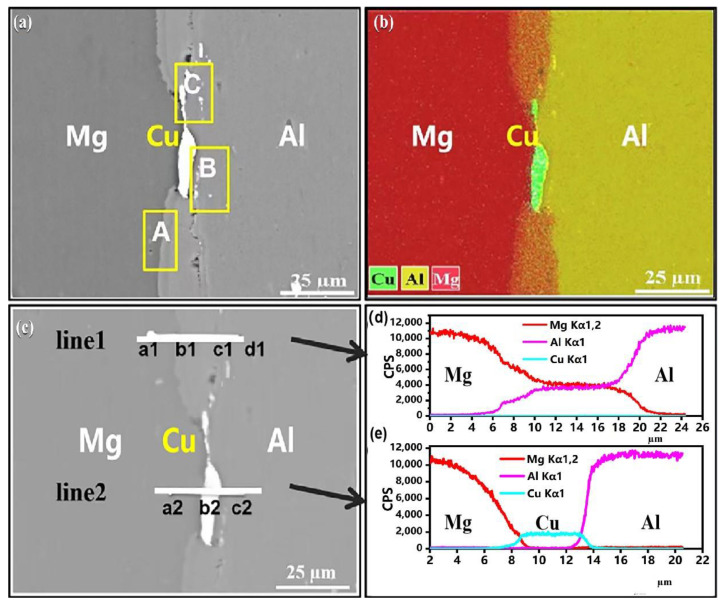
SEM and EDS images of TC351 interface: (**a**) SEM image of Mg/Al interface, (**b**) EDS map for Mg/Al interface, (**c**) EDS line sweep of the interface, (**d**) EDS results of line1, (**e**) EDS results of line2.

**Figure 7 materials-18-00655-f007:**
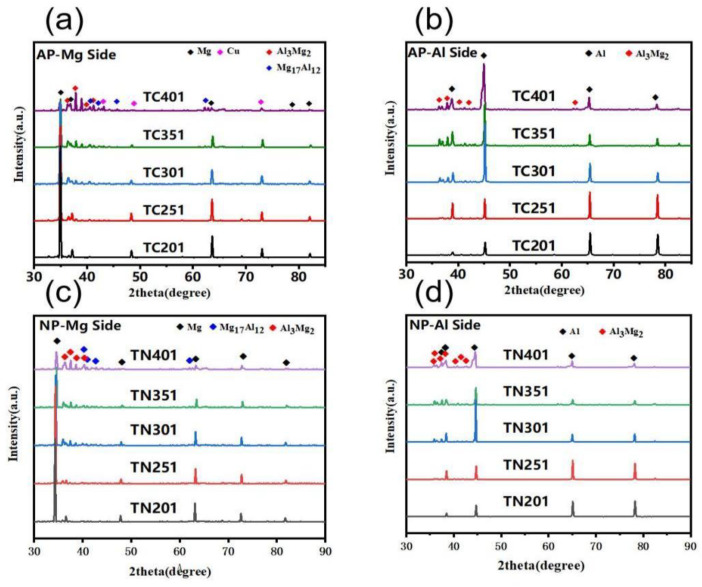
XRD patterns at different annealing temperatures. (**a**) Mg end in AP, (**b**) Al end in AP, (**c**) Mg end in NP, (**d**) Al end in NP.

**Figure 8 materials-18-00655-f008:**
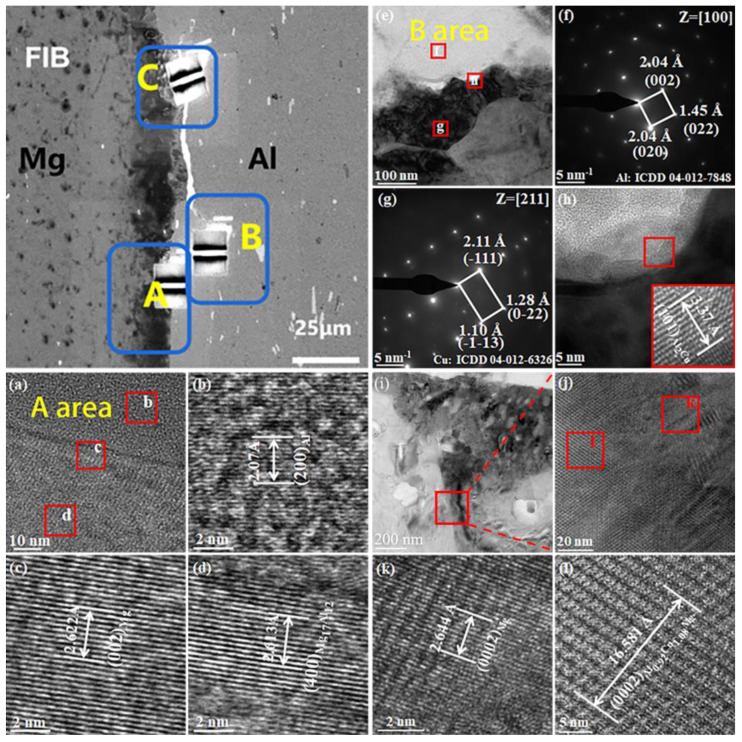
TEM analysis of the transition zone of the AP composite plate after 1 h of annealing at 350 °C, (**a**–**d**) TEM results in the A region adjacent to the Mg side, (**e**–**h**) TEM results in the B region adjacent to the Al side, (**i**–**l**) TEM results of the C region containing Cu powder.

**Figure 9 materials-18-00655-f009:**
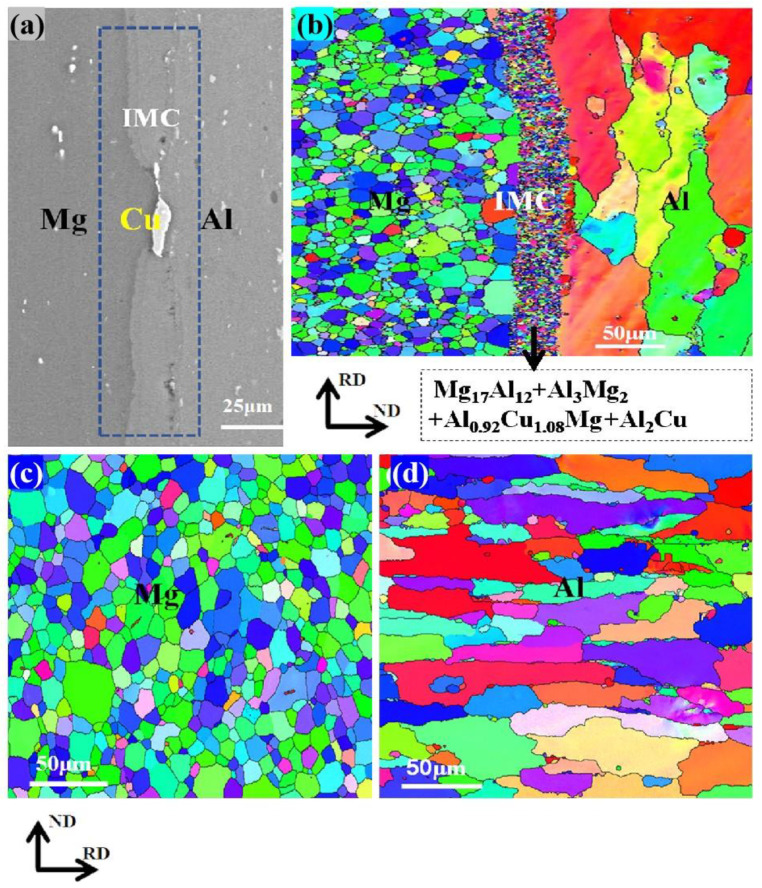
IPF diagrams of AP and NP: (**a**) Interfacial characterization of the AP composite plate after annealing at 350 °C for 1 h, (**b**) phase map from the (**a**), (**c**) IPF diagram of the Mg portion of the NP composite sheet; (**d**) IPF diagram of Al side of NP composite plate.

**Table 1 materials-18-00655-t001:** Chemical compositions of the experimental alloys (wt.%).

Material	Al	Mg	Zn	Zr	Si	Fe	Ni	Cu	Be
M21	-	Bal.	3.80	0.85	0	-	0.01	0.10	0.002
6061	Bal.	1.10	0.14	-	0.69	0.56	-	0.23	-

**Table 2 materials-18-00655-t002:** The annealing parameters and their associated labels.

Parameters	AP	TC201	TC251	TC301	TC351	TC401
	NP	TN201	TN251	TN301	TN351	TN401
Annealing temperature (°C)		200	250	300	350	400
Annealing time (h)		1	1	1	1	1

**Table 3 materials-18-00655-t003:** The measure of thickness in the interfacial diffusion layers of the AP and NP composite plates.

Style	Interfacial Diffusion Layer Thickness/µM				
	Sample Mark	TC251	TC301	TC351	TC401
**AP**	**AP-** **without Cu powder area**	5.6 ± 0.8	13.7 ± 1.7	18.5 ± 1.1	57.9 ± 0.8
	**AP-with Cu powder area**	1.0 ± 0.8	3.2 ± 0.9	1.87 ± 1.0	18.8 ± 2.5
**NP**	NP	**TN251**	**TN301**	**TN351**	**TN401**
		7.0 ± 0.4	17.4 ± 1.3	29.6 ± 1.6	61.2 ± 1.5

**Table 4 materials-18-00655-t004:** Determined growth velocities of IMCs at the Al/Mg interface.

Temperature/°C	D_(AP-Without Cu-Powder Area)_ m^2^/s	D_(AP-with Cu-Powder Area)_ m^2^/s	D_(NP)_ m^2^/s
250	1.25 × 10^−14^	2.78 × 10^−16^	1.36 × 10^−15^
300	5.06 × 10^−14^	2.84 × 10^−15^	8.41 × 10^−14^
350	2.42 × 10^−13^	1.00 × 10^−15^	2.43 × 10^−13^
400	9.93 × 10^−13^	9.81 × 10^−14^	1.04 × 10^−12^

**Table 5 materials-18-00655-t005:** EDS outcomes at the corresponding points within group line1 and line2.

Point No.	Chemical Compositions (at.%)			Probable Ingredient
	Mg	Al	Cu	
a1	94.8	5.2	0	Mg matrix
b1	56.1	43.9	0	Mg_17_Al_12_
c1	39.3	60.7	0	Al_3_Mg_2_
d1	1.9	98.1	0	Al matrix
a2	87.0	10.8	2.2	Mg-Al-Cu intermetallic compounds
b2	3.5	1.0	95.5	Cu
c2	4.0	95.4	0.6	Al matrix

## Data Availability

The original contributions presented in this study are included in the article. Further inquiries can be directed to the corresponding author.

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
