# Peer review of "The Effects of Cu Powder on the Interface Microstructure Evolution of Hot-Rolled Al 6061/Mg M21/Al 6061 Composite Plates During Annealing"

_materials, 2025, doi:10.3390/ma18030655_

Round 1

Reviewer 1 Report

Comments and Suggestions for Authors

The paper is interesting and can be published after a major revision:

Main points:

- how much Cu powder was placed between the plates and what is the surface density of these powders?

- For the XRD analysis, it would be useful to add a quantitative analysis of the phases present in the various locations, otherwise it is too speculative

Some minor points:

- in the abstract there is no space between sentences

- define IMC in line 18 of the abstract (at the moment it is defined on line 23); IMC acronym is defined at least 4 times afterwards

- Punctuation is wrong in many sites

- The quality of the inset figure in Fig 3 is not sufficient 

- Table 3: there is one entry which is "μm±1.0", please amend

- Table 3: the title read as "/μM". It should be "/μm"

- the labels inside Figures 6e and f cannot be read

- In the caption of Figure 6, explain what are the A, B, and C rectangles in Figure 6a

- The caption of Figure 8 needs to be more explicative

Author Response

1、how much Cu powder was placed between the plates and what is the surface density of these powders?

Response:

In the experimental procedure, the coverage rate of Cu powder on the slab was 80%, a parameter that has been adjusted during the course of the experiment as documented and highlighted them in red at the respective positions in the revised manuscript.

Commercial M21 Mg and 6061Al (200mm×100mm×1.0 mm) plates and Cu powder (~75 μm) were used in the experiment. The coverage rate of Cu powder on the slab was 80%. 

2、For the XRD analysis, it would be useful to add a quantitative analysis of the phases present in the various locations, otherwise it is too speculative

Response:

Thanks for reviewer’s comment.The content of intermetallic compounds at the interface of the composite plate utilized in this study is minimal and cannot be precisely detected by XRD. Consequently, XRD was employed to analyze the cross-sectional area instead of the interface position to ascertain the phase composition at the interface. The quantitative phase analysis in this study is more accurately determined by the thickness of each layer within the interface structure. As the variations in the thickness of the interfacial phases with temperature are detailed in Table 3, XRD was not utilized for quantitative analysis.

3、in the abstract there is no space between sentences

Response:

   Thanks for reviewer’s comment. The revised manuscript has been revised and adjusted to conform to the journal's formatting requirements.

4、define IMC in line 18 of the abstract (at the moment it is defined on line 23); IMC acronym is defined at least 4 times afterwards

Response:

   Thank you for pointing this out. We have revised and fine-tuned the repeatedly defined IMCs in the manuscript.. and highlighted them in red at the respective positions in the revised manuscript. For example:

line24:diffusion rate of IMCs in the Cu powder-containing region of the composite plate is significantly

5、Punctuation is wrong in many sites

Response:Thanks for reviewer’s comment.We made corrections to the punctuation throughout the article as needed, and made uniform changes where necessary, highlighting the corrected text in red, such as at line 55,the first sentence should conclude with a period, and the comma has been removed.As detailed below:

brittle intermetallic compounds. Preventing their rapid growth and ensuring a dispersed

6、The quality of the inset figure in Fig 3 is not sufficient

Response:

Thanks for reviewer’s comment.We have modified and adjusted Fig 3 accordingly and marked it in red on page 5 of the paper. The details are in word.

7、Table 3: there is one entry which is "μm±0", please amend

Response:

Thanks for reviewer’s comment.We have modified in Table3 accordingly and marked it in red at the respective positions in the revised manuscript.

8、Table 3: the title read as "/μM". It should be "/μm"

Response:

Thanks for reviewer’s comment. We have modified in Table3 accordingly and marked it in red at the respective positions in the revised manuscript.

9、the labels inside Figures 6e and f cannot be read

Response:

Thanks for reviewer’s comment.We have modified in Figures 6 accordingly and marked it in red at the respective positions in the revised manuscript.

10、In the caption of Figure 6, explain what are the A, B, and C rectangles in Figure 6a

Response:

Thank you for pointing this out.A, B, and C in Fig6 represent the diffusion layer near the Mg side, near the Al side, containing Cu powder. We have marked it in red at the respective positions in the revised manuscript.The details are as follows:

Fig 6, area A refers to the diffusion layer located near the Mg side, area B refers to the diffusion layer located near the Al side, and area C refers to the diffusion layer containing Cu powder.

11、The caption of Figure 8 needs to be more explicative

Response:

Thank you for your valuable comments. We have added the detailed title content in Fig8 and marked it in red at the respective positions in the revised manuscript. The details are as follows:

Figure 8. TEM analysis of the interface of the AP composite plate after 1 hour of annealing at 350°C, (a-d) TEM results in the A region adjacent to the Mg side, (e-h) TEM results in the B region adjacent to the Al side, (i-I) TEM results of the C region containing Cu powder.

Reviewer 2 Report

Comments and Suggestions for Authors

6061/M21/6061 composite sheets with Cu powder were studied using different experimental techniques. The manuscript is well written and the results are interesting. Some issues should be addressed before considering for publication.

1. In the Abstract, what is the meaning of IMC? The Abstract should not include abbreviations.
2. "The chemical composition of Al plate and Mg plate 93 was shown in Table 1." -> "The chemical composition of Al plate and Mg plate 93 is shown in Table 1."
3. Can the authors provide some insights into why different mechanisms are observed in Fig. 3?
4. Please check the English grammar. There are a few errors throughout the text.

Author Response

1、In the Abstract, what is the meaning of IMC? The Abstract should not include abbreviations.

Response:

  Intermetallic compounds, commonly abbreviated as IMCs,which have been modified in the revised manuscript in the abstract section on line 18 and marked it in red at the respective positions.Due to the word count limitation in the abstract, we utilized the abbreviated form.The details are as follows:

intermetallic compounds (IMCs) at the interface of Al-Mg composite plates

2、"The chemical composition of Al plate and Mg plate 93 was shown in Table 1." -> "The chemical composition of Al plate and Mg plate 93 is shown in Table 1."

Response:

Thank you for pointing this out.We have provided the necessary explanation.This refers to the chemical composition of Al plate 6061 and Mg plate M21, which we revised at the corresponding position in the revised manuscript.The details are as follows:

The chemical composition of Al plate 6061 and Mg plate M21 was shown in Table 1.

3、Can the authors provide some insights into why different mechanisms are observed in Fig. 3

Response:

Thanks for reviewer’s comment.We have revised and expanded the mechanism of interface morphology changes presented in Fig.3 and we marked it in red at the respective positions in the revised manuscript.The details are as follows:

“At an annealing temperature of 400°C, the diffusion of IMCs at the interface of the AP composite plate is impeded by the presence of Cu powder, resulting in a discontinuous diffusion state for Mg-Al IMCs that circumvents the Cu powder region.This is because as the temperature increases, the atomic motion of the atoms usually increases, promoting diffusion, but the presence of copper powder prevents or slows down the mutual diffusion between aluminum and magnesium. The copper powder may have changed the formation and growth rate of the metal inter-compounds (IMCs), resulting in an uneven diffusion state at the interface.In areas where Cu powder is present, the thickness of the diffusion layer at the Mg-Al interface is reduced, with a few holes observed on the IMCs and uneven distribution of Cu within a white strip morphology. Additionally, some Cu particles are spalling off, leading to irregular holes on IMCs in regions containing Cu powder.This is due to the differential diffusion rates between the magnesium and aluminum atoms.Previous studies have shown that [22] aluminum atoms diffuse much faster than Mg atoms. In the annealing process, Al atoms and Mg atoms diffuse together, forming an intermetallic phase at the Al/Mg interface. The difference in the diffusion rate of Al atoms and Mg atoms leads to the loss of atoms near the Al side, resulting in Kirkendell holes.”

4、Please check the English grammar. There are a few errors throughout the text.

Response:

Thank you for pointing this out.We have made corrections to the grammar errors throughout the article, for example:

We have modified this sentence “In  B area,Al-Cu interface,diffraction analysis showed that Al2Cu phase existed, as shown in Fig.8(d).” into the following sentence"In the B area, at the Al-Cu interface, diffraction analysis showed that the Al2Cu phase existed, as shown in Figure 8(d)."

Added "the" before "B area" to specify that we are talking about a particular area.

Added "at" before "Al-Cu interface" to clarify the location where the analysis was conducted.

Round 2

Reviewer 1 Report

Comments and Suggestions for Authors

The authors have sufficiently improved the manuscript.

Author Response

Dear reviewer:

Thank you for your time and effort in reviewing our manuscript

 (materials-3363040).  Your comments have been very helpful in improving the quality of our work.

 With kind regards!

 Yours sincerely,

 Xianquan Jiang

Reviewer 2 Report

Comments and Suggestions for Authors

The manuscript is ready for publication.

Author Response

(The authors gave the same response as above.)
